# ATTRACTION-REPULSION ACTOR-CRITIC FOR CONTINUOUS CONTROL REINFORCEMENT LEARNING

## ABSTRACT

In reinforcement learning, robotic control tasks are often useful for understanding how agents perform in environments with deceptive rewards where the agent can easily become trapped into suboptimal solutions. One way to avoid these local optima is to use a population of agents to ensure coverage of the policy space (a form of *exploration*), yet learning a population with the "best" coverage is still an open problem. In this work, we present a novel approach to population-based RL in continuous control that leverages properties of normalizing flows to perform attractive and repulsive operations between current members of the population and previously observed policies. Empirical results on the MuJoCo suite demonstrate a high performance gain for our algorithm compared to prior work, including Soft-Actor Critic (SAC).

## 1 INTRODUCTION

Many important reinforcement learning (RL) tasks, such as those in robotics and self-driving cars, are challenging due to large action and state spaces (Lee et al., 2018). In particular, environments with large continuous action spaces are prone to *deceptive* rewards, i.e. fall into local optima in learning (Conti et al., 2018). Applying traditional policy optimization algorithms to these domains often leads to locally optimal, yet globally sub-optimal policies. The agent should then explore the reward landscape more thoroughly in order to avoid falling into these local optima.

Not all RL domains that require exploration are suitable for understanding how to train agents that are robust to deceptive rewards. For example, Montezuma's Revenge, a game in the Atari Learning Environment (Bellemare et al., 2013), has *sparse rewards*; algorithms that perform the best on this task encourage exploration by providing a denser *intrinsic reward* to the agent to encourage exploration (Tang et al., 2017). On the other hand, many robotic control problems, such as those found in MuJoCo (Todorov et al., 2012), provide the agent with a dense reward signal, yet their high-dimensional action spaces induce a multimodal, often *deceptive*, reward landscape. For example, in the biped environments, coordinating both arms and legs is crucial for performing well on even simple tasks such as forward motion. However, simply learning to maximize the reward can be detrimental across training: agents will tend to run and fall further away from the start point rather than discovering stable and efficient walking motion. In this setting, *exploration* serves to provide a more reliable learning signal for the agent by covering more different types of actions during learning.

One way to maximize action space coverage is the maximum entropy RL framework (Ziebart, 2010), which prevents variance collapse by adding a policy entropy auxiliary objective. One such prominent algorithm, Soft Actor-Critic (SAC,Haarnoja et al. (2018)), has been shown to excel in large continuous action spaces. To further improve on exploration properties of SAC, one can maintain a population of agents that cover non-identical sections of the policy space. To prevent premature convergence, a diversity-preserving mechanism is typically put in place; balancing the objective and the diversity term becomes key to converging to a global optimum (Hong et al., 2018). This paper studies a particular family of population-based exploration methods, which conduct coordinated local search in the policy space. Prior work on population-based strategies improves performance on robotic control domains through stochastic perturbation on a single actor's parameter (Pourchot & Sigaud, 2019) or a set of actor's parameters (Conti et al., 2018; Khadka & Tumer, 2018; Liu et al., 2017). We hypothesize that exploring directly in the policy space will be more effective than perturbing

the parameters of the policy, as the latter does not guarantee diversity (i.e., different neural network parameterizations can approximately represent the same function).

Given a population of RL agents, we enforce local exploration using an Attraction-Repulsion (AR) mechanism. The later consists in adding an auxiliary loss to encourage pairwise attraction or repulsion between members of a population, as measured by a divergence term. We make use of the Kullback-Leibler (KL) divergence because of its desirable statistical properties and its easiness of computation. However, naively maximizing the KL term between two Gaussian policies can be detrimental (e.g. drives both means apart). Because of this, we parametrize the policy with a general family of distributions called Normalizing Flows (NFs, Rezende & Mohamed, 2015); this modification allows to improve upon AR+Gaussian (see Appendix Figure 6). NFs are shown to improve the expressivity of the policies using invertible mappings while maintaining entropy guarantees (Mazoure et al., 2019; Tang & Agrawal, 2018). Nonlinear density estimators have also been previously used for deep RL problems in contexts of distributional RL (Doan et al., 2018) and reward shaping (Tang et al., 2017). The AR objective blends particularly well with SAC, since computing the KL requires stochastic policies with tractable densities for each agent.

## 2 PRELIMINARIES

We first formalize the RL setting in a Markov decision process (MDP). A discrete-time, finite-horizon, MDP (Bellman, 1957; Puterman, 2014) is described by a state space $\mathcal{S}$, an action space $\mathcal{A}$, a transition function $\mathcal{P} : \mathcal{S} \times \mathcal{A} \times \mathcal{S} \mapsto \mathbb{R}^+$, and a reward function $r : \mathcal{S} \times \mathcal{A} \mapsto \mathbb{R}$.[1] On each round $t$, an agent interacting with this MDP observes the current state $s_t \in \mathcal{S}$, selects an action $a_t \in \mathcal{A}$, and observes a reward $r(s_t, a_t) \in \mathbb{R}$ upon transitioning to a new state $s_{t+1} \sim \mathcal{P}(s_t, a_t)$. Let $\gamma \in [0, 1]$ be a discount factor. The goal of an agent evolving in a discounted MDP is to learn a policy $\pi : \mathcal{S} \times \mathcal{A} \mapsto [0, 1]$ such as taking action $a_t \sim \pi(\cdot|s_t)$ would maximize the expected sum of discounted returns,

$$V^\pi(s) = \mathbb{E}_\pi \left[ \sum_{t=0}^\infty \gamma^t r(s_t, a_t) | s_0 = s \right].$$

In the following, we use $\rho_\pi$ to denote the trajectory distribution induced by following policy $\pi$. If $\mathcal{S}$ or $\mathcal{A}$ are vector spaces, action and space vectors are respectively denoted by $\mathbf{a}$ and $\mathbf{s}$.

### 2.1 DISCOVERING NEW SOLUTIONS THROUGH POPULATION-BASED ATTRACTION-REPULSION

Consider evolving a population of $M$ agents, also called *individuals*, $\{\pi_{\theta_m}\}_{m=1}^M$, each agent corresponding to a policy with its own parameters. In order to discover new solutions, we aim to generate agents that can mimic some target policy while following a path different from those of other policies.

Let $\mathcal{G}$ denote an archive of policies encountered in previous generations of the population. A natural way of enforcing $\pi$ to be different from or similar to the policies contained in $\mathcal{G}$ is by augmenting the loss of the agent with an Attraction-Repulsion (AR) term:

$$\mathcal{L}_{\text{AR}} = - \mathbb{E}_{\pi' \sim \mathcal{G}} \left[ \beta_{\pi'} D_{\text{KL}}[\pi||\pi'] \right], \tag{1}$$

where $\pi'$ is an archived policy and $\beta_{\pi'}$ is a coefficient weighting the relative importance of the Kullback-Leibler (KL) divergence between $\pi$ and $\pi'$, which we will choose to be a function of the average reward (see Sec. 3.2 below). Intuitively, Eq. 1 adds to the agent objective a weighted average distance between the current and the archived policies. For $\beta_{\pi'} \geq 0$, the agent tends to move away from the archived policy's behavior (i.e. *repulsion*, see Figure 1) a). On the other hand, $\beta_{\pi'} < 0$ encourages the agent $\pi$ to imitate $\pi'$ (i.e. *attraction*).

**Requirements for AR** In order for agents within a population to be trained using the proposed AR-based loss (Eq. 1), we have the following requirements:

1. Their policies should be stochastic, so that the KL-divergence between two policies is well-defined.

---

[1] $\mathcal{A}$ and $\mathcal{S}$ can be either discrete or continuous.

2. Their policies should have tractable distributions, so that the KL-divergence can be computed easily, either with closed-form solution or Monte Carlo estimation.

Several RL algorithms enjoy such properties (Haarnoja et al., 2018; Schulman et al., 2015; 2017). In particular, the soft actor-critic (SAC, Haarnoja et al., 2018) is a straightforward choice, as it currently outperforms other candidates and is off-policy, thus maintains a single critic shared among all agents (instead of one critic per agent), which reduces computation costs.

## 2.2 SOFT ACTOR-CRITIC

SAC (Haarnoja et al., 2018) is an off-policy learning algorithm which finds the information projection of the Boltzmann Q-function onto the set of diagonal Gaussian policies $\Pi$:

$$\pi = \arg\min_{\pi' \in \Pi} D_{KL}\left(\pi'(.|\mathbf{s}_t)\middle\|\frac{\exp\left(\frac{1}{\alpha}Q^{\pi_{\text{old}}}(\mathbf{s}_t, .)\right)}{Z^{\pi_{\text{old}}}(\mathbf{s}_t)}\right),$$

where $\alpha \in (0, 1)$ controls the temperature, i.e. the peakedness of the distribution. The policy $\pi$, critic $Q$, and value function $V$ are optimized according to the following loss functions:

$$\mathcal{L}_{\pi,\text{SAC}} = \mathbb{E}_{\mathbf{s}_t \sim \mathcal{B}}[\mathbb{E}_{\mathbf{a}_t \sim \pi}[\alpha \log \pi(\mathbf{a}_t|\mathbf{s}_t) - Q(\mathbf{s}_t, \mathbf{a}_t)]] \tag{2}$$

$$\mathcal{L}_Q = \mathbb{E}_{(s,a,r,s') \sim \mathcal{B}}\left[\{Q(s,a) - (r + \gamma V_\nu^\pi(s'))\}^2\right] \tag{3}$$

$$\mathcal{L}_V = \mathbb{E}_{\mathbf{s}_t \sim \mathcal{D}}\left[\frac{1}{2}\left\{V_\nu^\pi(\mathbf{s}_t) - \mathbb{E}_{\mathbf{a}_t \sim \pi}[Q(\mathbf{s}_t, \mathbf{a}_t) - \alpha \log \pi(\mathbf{a}_t|\mathbf{s}_t)]\right\}^2\right], \tag{4}$$

where $\mathcal{B}$ is the replay buffer. The policy used in SAC as introduced in Haarnoja et al. (2018) is Gaussian, which is both stochastic and tractable, thus compatible with our AR loss function in Eq. 1. Together with the AR loss in Eq. 1, the final policy loss becomes:

$$\mathcal{L}_\pi = \mathcal{L}_{\pi,\text{SAC}} + \mathcal{L}_{\text{AR}} \tag{5}$$

However, Gaussian policies are arguably of limited expressibility; we can improve on the family of policy distributions without sacrificing qualities necessary for AR or SAC by using Normalizing Flows (NFs, Rezende & Mohamed, 2015).

## 2.3 NORMALIZING FLOWS

NFs (Rezende & Mohamed, 2015) were introduced as a means of transforming simple distributions into more complex distributions using learnable and invertible functions. Given a random variable $\mathbf{z}_0$ with density $q_0$, they define a set of differentiable and invertible functions, $\{f_i\}_{i=1}^N$, which generate a sequence of $d$-dimensional random variables, $\{\mathbf{z}_i\}_{i=1}^N$.

Because SAC uses explicit, yet simple parametric policies, NFs can be used to transform the SAC policy into a richer one (e.g., multimodal) without risk loss of information. For example, Mazoure et al. (2019) enhanced SAC using a family of radial contractions around a point $\mathbf{z}_0 \in \mathbb{R}^d$,

$$f(\mathbf{z}) = \mathbf{z} + \frac{\beta}{\alpha + ||\mathbf{z} - \mathbf{z}_0||_2}(\mathbf{z} - \mathbf{z}_0) \tag{6}$$

for $\alpha \in \mathbb{R}^+$ and $\beta \in \mathbb{R}$. This results in a rich set of policies comprised of an initial noise sample $\mathbf{a}_0$, a state-noise embedding $h_\theta(\mathbf{a}_0, \mathbf{s}_t)$, and a flow $\{f_{\phi_i}\}_{i=1}^N$ of arbitrary length $N$, parameterized by $\phi = \{\phi_i\}_{i=1}^N$. Sampling from the policy $\pi_{\phi,\theta}(\mathbf{a}_t|\mathbf{s}_t)$ can be described by the following set of equations:

$$\begin{aligned} \mathbf{a}_0 &\sim \mathcal{N}(0, \mathbf{I}); \\ \mathbf{z} &= h_\theta(\mathbf{a}_0, \mathbf{s}_t); \\ \mathbf{a}_t &= f_{\phi_N} \circ f_{\phi_{N-1}} \circ ... \circ f_{\phi_1}(\mathbf{z}), \end{aligned} \tag{7}$$

where $h_\theta = \mathbf{a}_0 \sigma \mathbf{I} + \mu(\mathbf{s}_t)$ depends on the state and the noise variance $\sigma > 0$. Different SAC policies can thus be crafted by parameterizing their NFs layers.

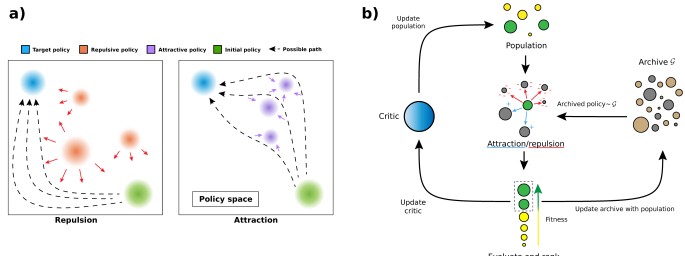

Figure 1: **a)** Augmenting the loss function with AR constraints allows an agent to reach a target policy by following different paths. Attractive and Repulsive policies represent any other agent's policy. **b)** General flow of the proposed ARAC strategy.

# 3 ARAC: ATTRACTION-REPULSION ACTOR-CRITIC

We now detail the general procedure for training a population of agents using the proposed diversity-seeking AR mechanism. More specifically, we consider here SAC agents enhanced with NFs (Mazoure et al., 2019). Figure 1 displays the general flow of the procedure. Algorithm 1 (Appendix) provides the pseudo-code of the proposed ARAC strategy, where sub-procedures for rollout and archive update can be found in the Appendix.

**Overview**    ARAC works by evolving a population of $M$ SAC agents $\{\pi_{\phi,\theta}^m\}_{m=1}^M$ with radial NFs policies (Eq. 7) and shared critic $Q_\omega$, and by maintaining an archive of policies encountered in previous generations of the population. After performing $T$ steps per agent on the environment (Alg. 1 L8-12), individuals are evaluated by performing $R$ rollouts[2] on the environment (Alg. 1 L26-28). This allows to identify the top-$K$ best agents  (Alg. 1 L29), also called *elites*, which will be used to update the critic as they provide the most meaningful feedback (Alg. 1 L13-17). The archive is finally updated in a diversity-seeking fashion using the current population (Alg. 1 L30).

The core component of the proposed approach lies within the update of the agents (Alg. 1 L18-25). During this phase, elite individuals are updated using AR operations w.r.t. policies sampled from the archive (Eq. 5), whereas non-elites are updated regularly (Eq. 2).

## 3.1 ENHANCING DIVERSITY IN THE ARCHIVE

Throughout the training process, we maintain an archive $\mathcal{G}$ of maximum capacity $G$, which contains some previously encountered policies. The process goes as follow: until reaching full capacity, the archive saves a copy of the parameters of every individual in the population after the evaluation step. However, by naively adding all individuals as if the archive were just a heap, the archive could end up filled with policies leading to similar rewards, which would result in a loss of diversity (Mauldin, 1984). We mitigate this issue by keeping track of two fitness clusters (low and high) using the partition formed by running a $k$-means algorithm on the fitness value. Hence, when $|\mathcal{G}| = G$ is reached and a new individual is added to the archive, it randomly replaces an archived policy from its respective cluster. This approach, also known as *niching*, has proved itself effective at maintaining high diversity levels (Gupta & Ghafir, 2012; Mahfoud, 1995).

## 3.2 DISCOVERING NEW POLICIES THROUGH ATTRACTION-REPULSION

The crux of this work lies in the explicit search for diversity in the policy space achieved using the AR mechanism. Since the KL between two base policies (i.e. input of the first flow layer) can be trivially maximized by driving their means apart, we apply attraction-repulsion only on the flow layers, while holding the mean of the base policy constant. This ensures that the KL term doesn't depend on the difference in means and hence controls the magnitude of the AR mechanism. Every time the AR operator is applied (Alg. 1 L20-21), $n$ policies are sampled from the archive and are used for estimating the AR loss (Eq. 1). As in Hong et al. (2018), we consider two possible strategies

---

[2]These steps can be performed in parallel.

to dictate the value of $\beta_{\pi'}$ coefficients for policies $\pi' \sim \mathcal{G}$:

$$\beta_{\pi'} = -\left[2\left(\frac{f(\pi') - f_{min}}{f_{max} - f_{min}} - 1\right)\right] \qquad \text{(proactive)} \qquad (8)$$

$$\beta_{\pi'} = 1 - \frac{f(\pi') - f_{min}}{f_{max} - f_{min}} \qquad \text{(reactive)} \qquad (9)$$

where $f(\pi)^3$ represents the fitness function of policy $\pi$ (average reward in our case), and $f_{min}$ and $f_{max}$ are estimated based on the $n$ sampled archived policies. The *proactive* strategy aims to mimic high reward archived policies, while the *reactive* strategy is more cautious, only repulsing away the current policy from low fitness archived policies. Using this approach, the current agent policy will be attracted to some sampled policies ($\beta_{\pi'} < 0$) and will be repulsed from others ($\beta_{\pi'} \geq 0$) in a more or less aggressive way, depending on the strategy.

Unlike Hong et al. (2018) who applied proactive and reactive strategies on policies up to 5 timesteps back, we maintain an archive consisting of two clusters seen so far: policies with low and high fitness, respectively. Having this cluster allows to attract/repulse from a set of diverse agents, replacing high-reward policies by policies with similar performance. Indeed, without this process, elements of the archive would collapse on the most frequent policy, from which all agents would attract/repulse. To avoid performing AR against a single "average policy" , we separate low-reward and high-reward agents via clustering.

## 4    RELATED WORK

The challenges of exploration are well studied in the RL literature. Previously proposed approaches for overcoming hard exploration domains tend to either increase the capacity of the state-action value function (Gal & Ghahramani, 2016; Henderson et al., 2017) or the policy expressivity (Mazoure et al., 2019; Tang & Agrawal, 2018; Touati et al., 2018). This work rather tackles exploration from a diverse multi-agent perspective. Unlike prior population-based approaches for exploration (Conti et al., 2018; Khadka & Tumer, 2018; Pourchot & Sigaud, 2019), which seek diversity through the parameters space, we directly promote diversity in the policy space.

The current work was inspired by Hong et al. (2018), who relied on the KL divergence to attract/repulse from a set of previous policies to discover new solutions. However, in their work, the archive is time-based (they restrict themselves to the 5 most recent policies), while our archive is built following a diversity-seeking strategy (i.e., niching and policies come from multiple agents). Notably, ARAC is different of previously discussed works in that it explores the action space in multiple regions simultaneously, a property enforced through the AR mechanism.

The proposed approach bears some resemblance with Liu et al. (2017), who took advantage of a multi-agent framework in order to perform repulsion operations among agents using of similarity kernels between parameters of the agents. The AR mechanism gives rise to exploration through structured policy rather than randomized policy. This strategy has also been employed in multi-task learning (Gupta et al., 2018), where experience on previous tasks was used to explore on new tasks.

## 5    EXPERIMENTS

### 5.1    DIDACTIC EXAMPLE

Consider a 2-dimensional multi-armed bandit problem where the actions lie in the real square $[-6, 6]^2$. We illustrate the example of using a proactive strategy where a SAC agent with radial flows policy imitates a desirable (expert) policy while simultaneously repelling from a less desirable policy. The task consists in matching the expert's policy (blue density) while avoiding taking actions from a repulsive policy $\pi'$ (red). We illustrate the properties of radial flows in Figure 2 by increasing the number of flows (where 0 flow corresponds to a Gaussian distribution).

We observe that increasing the number of flows (bottom to top) leads to more complex policy's shapes and multimodality unlike the Gaussian policy which has its variance shrinked (the KL divergence

---

[3]We overload our notation f for both the normalizing flow and the fitness depending on the context

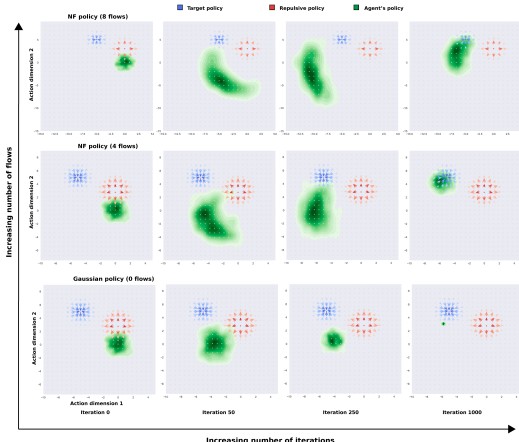

Figure 2: Agent trained to imitate a target while avoiding a repulsive policy using a proactive strategy. Increasing the number of flows leads to more complex policy's shape.

is proportional to the ratio of the two variances, hence maximizing it can lead to a reduction in the variance which can be detrimental for exploration purpose). Details are provided in Appendix.

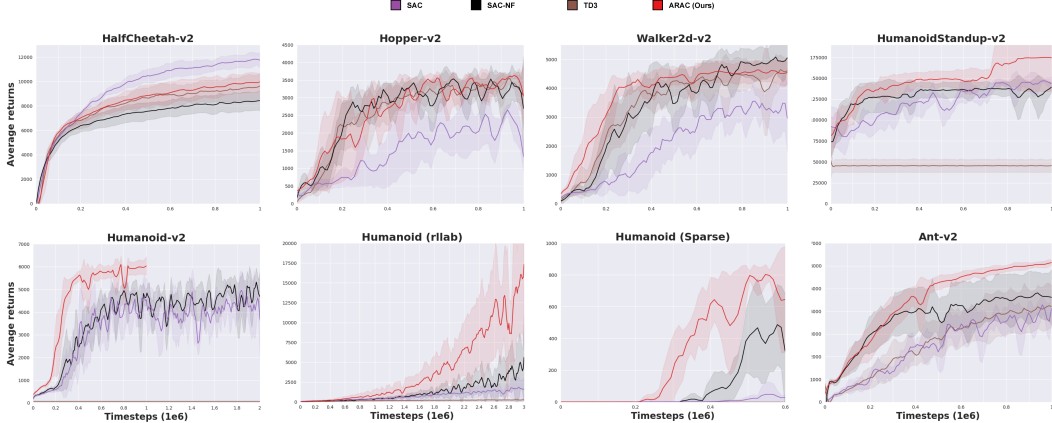

Figure 3: Average return and one standard deviation on 5 random seeds across 7 MuJoCo tasks for ARAC against single SAC agents (with and without NFs). Curves are smoothed using Savitzky-Golay filtering with window size of 7.

## 5.2 MuJoCo locomotion benchmarks

We now compare ARAC against the CEM-TD3 (Pourchot & Sigaud, 2019), ERL (Khadka & Tumer, 2018) and CERL (Khadka et al., 2019) multi-agent baselines on seven continuous control tasks from the MuJoco suite (Duan et al., 2016): `Ant-v2`, `HalfCheetah-v2`, `Humanoid-v2`, `HumanoidStandup-v2`, `Hopper-v2`, `Walker2d-v2` and `Humanoid (rllab)`. We also designed a sparse reward environment `SparseHumanoid-v2`. All algorithms are run over 1M time steps on each environment, except `Humanoid (rllab)` which gets 2M time steps and `SparseHumanoid-v2` on 0.6M time steps. We also include comparison against single-agent baselines.

ARAC performs $R = 10$ rollouts for evaluation steps every $10,000$ interaction steps with the environment. We consider a small population of $N = 5$ individuals with $K = 2$ as elites. Every SAC agent has one feedforward hidden layer of 256 units acting as state embedding, followed by a radial flow of length $\in \{3, 4\}$. A temperature of $\alpha = 0.05$ or $0.2$ is used across all the environments (See appendix for more details). AR operations are carried out by sampling uniformly $n = 5$ archived

policies from $\mathcal{G}$. Parameters details are provided in the Appendix (Table 4). All networks are trained with Adam optimizer (Kingma & Ba, 2015) using a learning rate of $3E^{-4}$. Baselines CEM-TD3[4], ERL[5], CERL[6] use the code contained in their respective repositories.

|  | ARAC | CEM-TD3 | CERL | ERL | SAC - NF | SAC | TD3 |
|---|---|---|---|---|---|---|---|
| Ant | **6044** | 4239 | 1639 | 1442 | 4912 | 4370 | 4372 |
| HC | 10 264 | 10 659 | 5703 | 6746 | 8429 | **11 900** | 9543 |
| Hopper | **3587** | **3655** | 2970 | 1149 | 3538 | 2794 | 3564 |
| Hu | **5965** | 212 | 4756 | 551 | 5506 | 5504 | 71 |
| Standup | **175 000** | 29 000 | 117 000 | 12 900 | 116 000 | 149 000 | 54 000 |
| Hu (rllab) | **14 230** | 1334 | 3340 | 57 | 5531 | 1963 | 286 |
| Walker2d | 4704 | 4710 | 4386 | 1107 | **5196** | 3783 | 4682 |
| Hu (Sparse) | **816** | 0 | 1.32 | 8.65 | 547 | 88 | 0 |

Table 1: Maximum average return after 1M (2M for `Humanoid (rllab)` and 600k for `SparseHumanoid-v2`) time steps 5 random seeds. Bold: best methods when the gap is less than 100 units. See appendix for average return with standard deviation. Environment short names: HC: `HalfCheetah-v2`, Hu: `Humanoid-v2`, Standup: `HumanoidStandup-v2`

Figure 4 displays the performance of all algorithms on three environments over time steps (see Appendix Figure 7 for all environments). Results are averaged over 5 random seeds. Table 1 reports the best observed reward for each method.

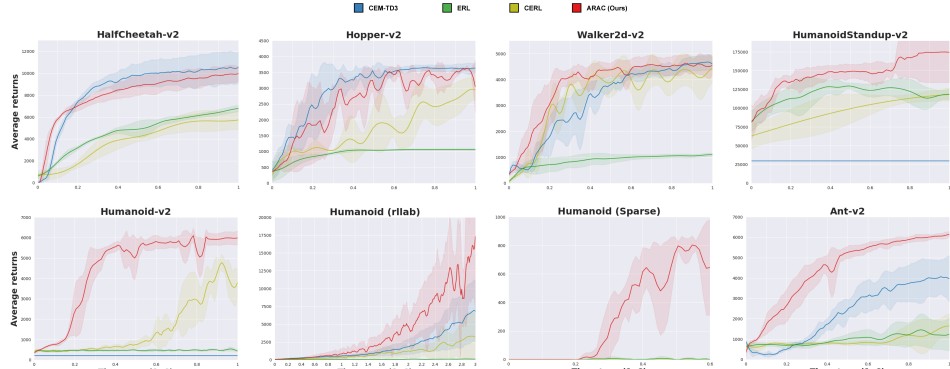

Figure 4: Average return and one standard deviation on 5 random seeds across 8 MuJoCo tasks. Curves are smoothed using Savitzky-Golay filtering with window size of 7.

**Small state space environments** `HalfCheetah-v2`, `Hopper-v2`, and `Walker2d-v2` are low-dimensional state space environments ($d \leq 17$). Except for `HalfCheetah-v2`, the proposed approach shows comparable results with its concurrent. Those results match the findings of (Plappert et al., 2018) that some environments with well-structured dynamics require little exploration. Full learning curves can be found in the Appendix.

**Deceptive reward and Large state space environments** `Humanoid-v2`, `HumanoidStandup-v2` and `Humanoid (rllab)` belong to bipedal environments with high-dimensional state space ($d = 376$ and $d = 147$), and are known to trap algorithms into suboptimal solutions. In addition to the legs, the agent also needs to control the arms, which may influence the walking way and hence induce deceptive rewards (Conti et al., 2018). Figure 4 shows the learning curves on MuJoCo tasks. We observe that ARAC beats both baselines in performance as well as in convergence rate.

`Ant-v2` is another high-dimensional state space environment ($d \geq 100$). In an unstable setup, a naive algorithm implementing an unbalanced fast walk could still generate high reward, the reward

---

[4]`https://github.com/apourchot/CEM-RL`
[5]`https://github.com/ShawK91/erl_paper_nips18`
[6]`https://github.com/IntelAI/cerl`

taking into account the distance from start, instead of learning to stand, stabilize, and walk (as expected).

**Sparse reward environment**   To test ARAC in a sparse reward environment, we created `SparseHumanoid-v2`. The dynamic is the same as Humanoid-v2 but rewards of +1 is granted only given is the center of mass of the agent is above a threshold (set to 0.6 unit in our case). The challenge not only lies in the sparse reward property but also on the complex body dynamic that can make the agent falling down and terminating the episode. As shown in Figure 4, ARAC is the only method that can achieve non zero performance. A comparison against single agent methods in the Appendix also shows better performance for ARAC.

**Sample efficiency compared with single agent methods**   Figure 3 (in Appendix) also shows that the sample efficiency of the population-based ARAC compares to a single SAC agent (with and without NFs) and other baselines methods (SAC, TD3). On `Humanoid-v2` and `Ant-v2` ARAC converges faster, reaching the 6k (4k, respectively) milestone performance after only 1M steps, while a single SAC agent requires 4M (3M, respectively) steps according to (Haarnoja et al., 2018). In general, ARAC achieves competitive results (no flat curves) and makes the most difference (faster convergence and better performance) in the biped environments.

**Attraction-repulsion ablation study**   To illustrate the impact of repulsive forces, we introduce a hyperparameter $\lambda$ in the overall loss (Eq. 5):

$$\mathcal{L}_{\theta,\phi,\lambda} = \mathcal{L}_{\theta,\phi,\mathrm{SAC}} + \lambda\mathcal{L}_{\phi,\mathrm{AR}} \tag{10}$$

We ran an ablation analysis on `Humanoid-v2` by varying that coefficient. For two random states, we sampled 500 actions from all agents and mapped these actions onto a two-dimensional space (via t-SNE). Appendix Figure 5 shows that without repulsion ($\lambda = 0$), actions from all agents are entangled, while repulsion ($\lambda > 0$) forces agents to behave differently and hence explore different regions of the action space.

The second ablation study is dedicated to highlight the differences between a Gaussian policy (similar to Hong et al. (2018) and an NF policy under AR operators. As one can observe in Figure 6, using a Gaussian policy deteriorates the solution as the repulsive KL term drives apart the means of agents and blows up/ shrinks the variance of the Gaussian policy. On the other hand, applying the AR term on the NF layers maximizes the KL conditioned on the mean and variance of both base policies, resulting in a solution which allows sufficient exploration. More details are provided in the Appendix.

Finally, through a toy example subject to AR, we characterize the policy's shape when increasing the number of the radial flow policy in Figure 2 (experimental setup in Appendix). Unlike the diagonal Gaussian policy (SAC) that has symmetry constraints, increasing the number of flows allows the radial policy to adopt more complex shapes (from bottom to top).

## 6   CONCLUSION

In this paper, we addressed the issue of RL domains with deceptive rewards by introducing a population-based search model for optimal policies using attraction-repulsion operators. Our method relies on powerful density estimators (normalizing flows), to let policies exploit the reward landscape under AR constraints. Our ablation studies showed that (1) the strength of AR and (2) the number of flows are the two factors which predominantly affect the shape of the policy. Selecting the correct AR coefficient is therefore important to obtain good performance, while at the same time preventing premature convergence.

Empirical results on the MuJoCo suite demonstrate high performance of the proposed method in most settings, including with sparse rewards. Moreover, in biped environments that are known to trap algorithms into suboptimal solutions, ARAC enjoys higher sample efficiency and better performance compared to its competitors which confirms our intuitions on using AR with normalizing flows. As future steps, borrowing from multi-objective optimization literature methods could allow one to combine other diversity metrics with the performance objective, to in turn improve the coverage of the solution space among the individuals by working with the corresponding Pareto front (Horn et al., 1994).

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

APPENDIX

REPRODUCIBILITY CHECKLIST

We follow the reproducibility checklist (Pineau, 2018) and point to relevant sections explaining them here.
For all algorithms presented, check if you include:

- **A clear description of the algorithm, see main paper and included codebase.** The proposed approach is completely described by Alg. 1 (main paper), 2 (Appendix), and 3 (Appendix). The proposed population-based method uses attraction-repulsion operators in order to enforce a better policy space coverage by different agents.

- **An analysis of the complexity (time, space, sample size) of the algorithm.** See Appendix Figure 7 and 3. Experimentally, we demonstrate improvement in sample complexity as discussed in our main paper. In term of computation time, the proposed method scales linearly with the population size if agents are evaluated sequentially (as presented in Alg. 1 for clarity). However, this as mentioned in the paper, can be parallelized. All our results are obtained using $M$ small network architectures with $1 \times 256$-units hidden layer followed by $f$ layers of $|A| + 2$ units each ($f$ being the number of radial flows and $|A|$ being the action space dimension).

- **A link to a downloadable source code, including all dependencies.** The code is included with the Appendix as a zip file; all dependencies can be installed using Python's package manager. Upon publication, the code would be available on Github.

For all figures and tables that present empirical results, check if you include:

- **A complete description of the data collection process, including sample size.** We use standard benchmarks provided in OpenAI Gym (Brockman et al., 2016).

- **A link to downloadable version of the dataset or simulation environment.** See: https://github.com/

- **An explanation of how samples were allocated for training / validation / testing.** We do not use a training-validation-test split, but instead report the mean performance (and one standard deviation) of the policy at evaluation time, openai/gym for OpenAI Gym benchmarks and https://www.roboti.us/index.html for MuJoCo suite. obtained with 5 random seeds.

- **An explanation of any data that were excluded.** We did not compare on easy environments (e.g. `Reacher-v2`) because all existing methods perform well on them. In that case, the improvement of our method upon baselines is incremental and not worth mentioning.

- **The exact number of evaluation runs.** 5 seeds for MuJoCo experiments, 1M, 2M or 3M environment steps depending on the domain.

- **A description of how experiments were run.** See Section 5 in the main paper and didactic example details in Appendix.

- **A clear definition of the specific measure or statistics used to report results.** Undiscounted returns across the whole episode are reported, and in turn averaged across 5 seeds.

- **Clearly defined error bars.** Confidence intervals and table values are always mean$\pm 1$ standard deviation over 5 seeds.

- **A description of results with central tendency (e.g. mean) and variation (e.g. stddev).** All results use the mean and standard deviation.

- **A description of the computing infrastructure used.** All runs used 1 CPU for all experiments (toy and MuJoCo) with 8Gb of memory.

IMPACT OF REPULSIVE FORCE

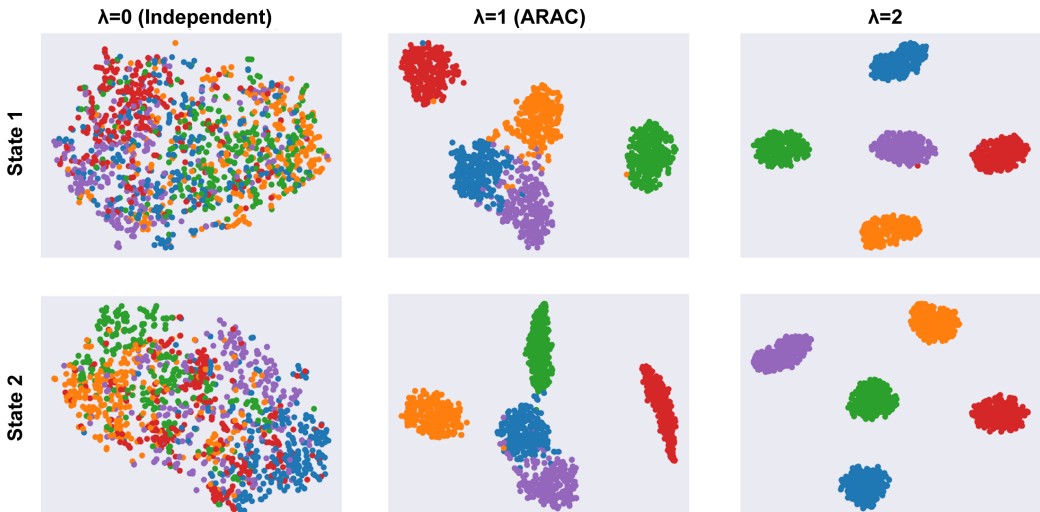

Figure 5: Mapping in two-dimension space (t-SNE) of agents' actions for two arbitrary states. Each color represents a different agent.

To illustrate the impact of the repulsive force coefficient $\lambda$, we ran an ablation analysis by varying that coefficient (recall that the overall loss function is $\mathcal{L}_\pi = \mathcal{L}_{\pi,\text{SAC}} + \lambda \mathcal{L}_{\text{AR}}$ where $\lambda = 1$ in our experiment).

For two random states, we sampled $500$ actions from all agents and mapped theses actions in a common 2-dimensional space (t-SNE).

As shown in the Figure above, policies trained without AR ($\lambda = 0$) result in entangled actions, while increasing the repulsive coefficient $\lambda$ forces agents to have different actions and hence explore different regions of the policy space. Note that due to the specific nature of t-SNE , the policies are shown as Gaussians in a lower-dimensional embedding, while it is not necessarily the case in the true space.

STABILIZING ATTRACTION-REPULSION WITH NORMALIZING FLOW

In this section, we illustrate the consequence of the AR operators with a Gaussian policy (as in Hong et al. (2018)) and our Normalizing flow policy for `Ant-v2`, `Humanoid-v2` and `HalfCheetah-v2`. As shown in the figure below, AR with Gaussian policies yield worse results. One reason is that the KL term drives apart the mean and variance of the Gaussian policy which deteriorates the main objective of maximizing the reward. On the other side, our method applies the AR only on the NF layers allows enough exploration by deviating sufficiently from the main objective function.

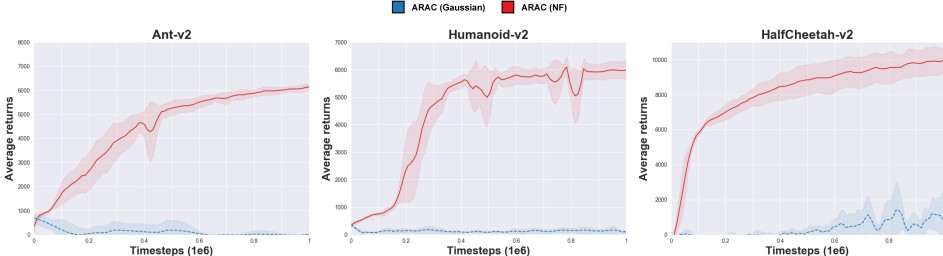

Figure 6: Comparison of ARAC agents using (1) AR with radial flows, (2) AR with only the base (Gaussian) policy and (3) no AR with radial flows.

COMPARING ARAC AGAINST BASELINES ON MUJOCO TASKS

Figure 7 shows the performance of ARAC and baselines (CEM-TD3, CERL and ERL) over time steps. Learning curves are averaged over 5 random seeds and displayed with one standard deviation. Evaluation is done every $10,000$ environment steps using 10 rollouts per agent. Overall, ARAC has reasonable performance on all tasks (no flat curves) and demonstrates high performance, especially in humanoid tasks.

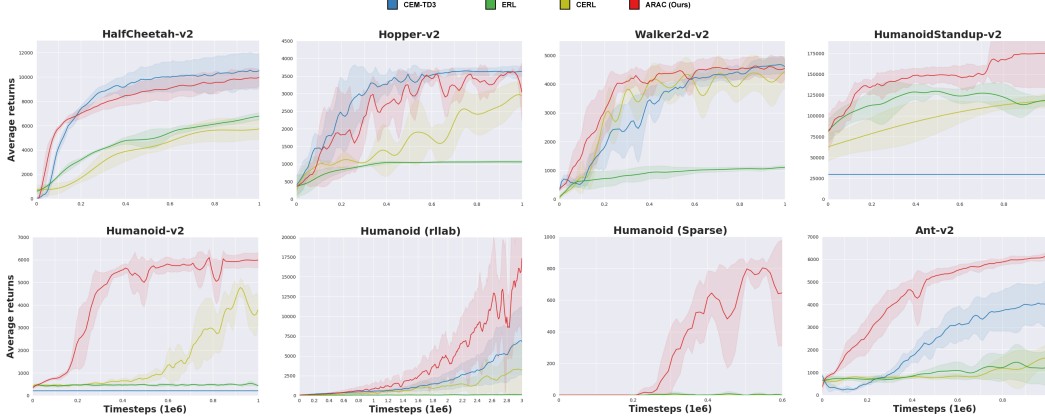

Figure 7: Average return and one standard deviation on 5 random seeds across 7 MuJoCo tasks for ARAC against baselines. Curves are smoothed using Savitzky-Golay filtering with window size of 7.

BENEFITS OF POPULATION-BASED STRATEGIES: ARAC AGAINST SINGLE AGENTS

In this section, we highlight the benefits of the proposed population-based strategy by comparing with single agents. Figure 3 shows the performance of ARAC against a single SAC agent (with and without normalizing flows). Learning curves are averaged over 5 random seeds and displayed with one standard deviation. Evaluation is done every $10,000$ environment steps using 10 rollouts per agent. We observe a high beneficial impact on the convergence rate as well as on the performance. ARAC outperforms single agents in almost all tasks (except for `HalfCheetah-v2` and `Walker-v2`) with large improvement. Note the high sample efficiency on humanoid environments (`Humanoid-v2` and `Humanoid (rllab)`), where it shows faster convergence in addition to better performance. Indeed, on `Humanoid (rllab)` a single SAC agent reaches the 4k milestone after 4M steps (Haarnoja et al., 2018) while ARAC achieves this performance in less than 2M steps. Also, in `SparseHumanoid-v2`, due to its better coordinated exploration, ARAC could find a good solution faster than SAC-NF.

OVERALL PERFORMANCES ON MUJOCO TASKS

| | ARAC | CEM-TD3 | CERL | ERL | SAC - NF | SAC | TD3 |
|---|---|---|---|---|---|---|---|
| Ant-v2 | **6,044 ± 216** | $4,239 \pm 1,048$ | $1,639 \pm 564$ | $1,442 \pm 819$ | $4,912 \pm 954$ | $4,370 \pm 173$ | $4,372 \pm 900$ |
| HalfCheetah-v2 | $10,264 \pm 271$ | $10,659 \pm 1,473$ | $5,703 \pm 831$ | $6,746 \pm 295$ | $8,429 \pm 818$ | **11,896 ± 574** | $9,543 \pm 978$ |
| Hopper-v2 | **3,587 ± 65** | **3,655 ± 82** | $2,970 \pm 341$ | $1,149 \pm 3$ | $3,538 \pm 108$ | $2,794 \pm 729$ | $3,564 \pm 114$ |
| Humanoid-v2 | **5,965 ± 51** | $212 \pm 1$ | $4,756 \pm 454$ | $551 \pm 60$ | $5,506 \pm 147$ | $5,504 \pm 116$ | $71 \pm 10$ |
| HumanoidStandup-v2 | **175k ± 38k** | $29k \pm 4k$ | $117k \pm 8k$ | $129k \pm 4k$ | $116k \pm 9k$ | $149k \pm 7k$ | $54k \pm 24k$ |
| Humanoid (rllab) | **14,234 ± 7251** | $1,334 \pm 551$ | $3,340 \pm 3,340$ | $57 \pm 17$ | $5,531 \pm 4,435$ | $1,963 \pm 1,384$ | $286 \pm 151$ |
| Walker2d-v2 | $4,704 \pm 261$ | $4,710 \pm 320$ | $4,3860 \pm 615$ | $1,107 \pm 60$ | **5,196 ± 527** | $3,783 \pm 366$ | $4,682 \pm 539$ |
| SparseHumanoid-v2 | **816 ± 20** | $0 \pm 0$ | $1.32 \pm 2.64$ | $8.65 \pm 15.90$ | $547 \pm 268$ | $88 \pm 159$ | $0 \pm 0$ |

Table 2: Maximum average return after 1M (2M for `Humanoid (rllab)` and 600k for `SparseHumanoid-v2`) time steps $\pm$ one standard deviation on 5 random seeds. Bold: best methods when the gap is less than 100 units.

| | ARAC | TRPO | PPO | Trust-PCL | Plappert et al. (2017) | Touati et al. (2018) | Hong et al. (2018) |
|---|---|---|---|---|---|---|---|
| HalfCheetah-v2 | **10,264** | $-15$ | $2,600$ | $2,200$ | $5,000$ | $7,700$ | $4,200$ |
| Walker-v2 | **4,764** | $2,400$ | $4,050$ | $400$ | $850$ | $500$ | N/A |
| Hopper-v2 | **3,588** | $600$ | $3,150$ | $280$ | $2,500$ | $400$ | N/A |
| Ant-v2 | **6,044** | $-76$ | $1,000$ | $1,500$ | N/A | N/A | N/A |
| Humanoid-v2 | **5,939** | $400$ | $400$ | N/A | N/A | N/A | $1,250$ |
| HumanoidStandup-v2 | **163,884** | $80,000$ | N/A | N/A | N/A | N/A | N/A |
| Humanoid (rllab) | **4,117** | $23$ | $200$ | N/A | N/A | N/A | N/A |

Table 3: Performance after 1M (except for rllab which is 2M) timesteps on 5 seeds. Values taken from their corresponding papers. N/A means the values were not available in the original paper.

EXPERIMENTAL PARAMETERS

Table 4 provides the hyperparameters of ARAC used to obtain results in the MuJoCo domains. The noise input for normalizing flows in SAC policies (see Sec. 2.3) is sampled from $\mathcal{N}(0, \sigma)$, where the variance $\sigma$ is a function of the state (either fixed at a given value or learned).

| ARAC parameters | | | | | | |
|---|---|---|---|---|---|---|
| | # flows | $\sigma$ | $G$ | $p$ | alpha | strategy |
| Ant-v2 | 3 | 0.2 | 10 | 1 | 0.2 | proactive |
| HalfCheetah-v2 | 4 | 0.4 | 20 | 2 | 0.2 | proactive |
| Hopper-v2 | 4 | 0.8 | 20 | 1 | 0.05 | proactive |
| Walker2d-v2 | 4 | 0.6 | 10 | 3 | 0.05 | proactive |
| Humanoid-v2 | 3 | 0.6 | 10 | 1 | 0.05 | reactive |
| HumanoidStandup-v2 | 3 | $\sigma$ | 20 | 1 | 0.2 | reactive |
| Humanoid (rllab) | 3 | $\sigma$ | 10 | 1 | 0.05 | proactive |
| SparseHumanoid-v2 | 2 | 0.6 | 20 | 1 | 0.2 | proactive |
| Adam Optimizer parameters | | | | | | |
| $\alpha_\gamma$ | $3.10^{-4}$ | | | | | |
| $\alpha_\omega$ | $3.10^{-4}$ | | | | | |
| $\alpha_\theta$ | $3.10^{-4}$ | | | | | |
| $\alpha_\phi$ | $3.10^{-4}$ | | | | | |
| Algorithm parameters | | | | | | |
| Batch size $m$ | 256 | | | | | |
| Buffer size $\mathcal{B}$ | $10^6$ | | | | | |
| Archive sample size $n$ | 5 | | | | | |

Table 4: ARAC parameters.

IMPACT OF NUMBER OF FLOWS ON THE POLICY SHAPE

We used a single SAC agent with different radial flows numbers and randomly initialized weights, starting with actions centered at $(0, 0)$. All flow parameters are $\ell_1$ regularized with hyperparameter 2. The agent is trained with the classical evidence lower bound (ELBO) objective augmented with the AR loss (Eq. 1), where the coefficient of the repulsive policy $\pi'$ is given by $\beta_t = \frac{10}{t+1}$. Fig. 8 shows how both the NF and learned variance Gaussian policies manage to recover the target policy. We see that NF takes advantage of its flexible parametrization to adjust its density and can show asymmetric properties unlike the Gaussian distribution. This indeed can have advantage in some non symmetric environment where the Gaussian policy would be trapped into a suboptimal behavior. Finally, increasing the number of flows (from bottom to top) can lead to more complex policy's shape.

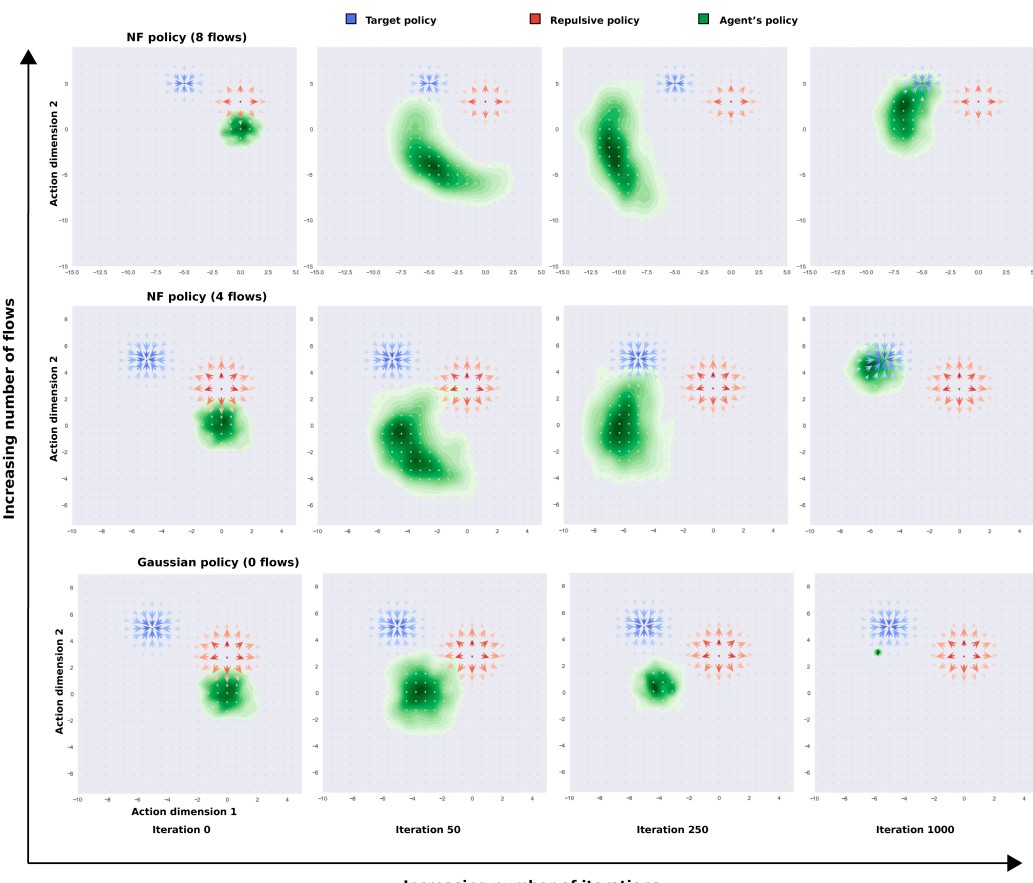

Figure 8: Single state didactic illustration of attraction-repulsion operators. Comparing behavior of NF policy against Gaussian policy with learned variance under a repulsive constraint.

## 6.1 VARIANCE OF FITNESS IN THE ARCHIVE

Due to the high computation time for behavioral-diversity baselines such as DIYAN, we propose to use the agent's fitness (i.e. undiscounted returns) as a candidate to repulse/attract from.

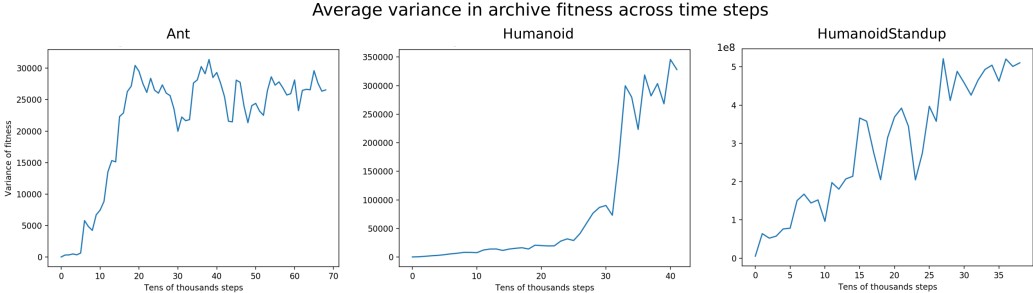

Figure ?? shows the variance of the archive across three MuJoCo domains: Ant, Humanoid and HumanoidStandup. As training progresses, the clustering approach allows to maintain a high variance in the archive, preventing mode collapse to a single, "average" fitness.

## 6.2 PSEUDO-CODE FOR ARAC

---

**Algorithm 1** ARAC: Attraction-Repulsion Actor-Critic

---

1: **Input:** population size $M$; number of elites $K$; maximum archive capacity $G$; archive sample size $n$; number of evaluation rollouts $R$; actor coefficient $p$; strategy (either proactive or reactive).

2: Initialize value function network $V_\nu$ and critic network $Q_\omega$
3: Initialize population of policy networks $\{\pi^m_{\phi,\theta}\}^M_{m=1}$
4: Initialize empty archive $\mathcal{G}$ and randomly assign $K$ individuals to top-$K$

5: total_step $\leftarrow 0$
6: **while** total_step $\leq$ max_step **do**
7:    step $\leftarrow 0$

8:    **for** agent $m = 1\ldots M$ **do**
9:      $(\_, \text{step } s) \leftarrow \texttt{rollout}(\pi^m, \text{with noise, over 1 episode})$          *Collect samples*
10:      step $\leftarrow$ step $+ s$
11:      total_step $\leftarrow$ total_step $+ s$
12:    **end for**

13:    $C = \text{step}/K$
14:    **for** policy $\pi^e$ in top-$K$ **do**
15:      Update critic with $\pi^e$ for $C$ mini-batches (Eq. 3)          *Update critic*
16:      Update value function (Eq. 4)
17:    **end for**

18:    **for** agent $m = 1\ldots M$ **do**
19:      **if** policy $\pi^m$ is in top-$K$ **then**
20:        Sample $n$ archived policies uniformly from $\mathcal{G}$
21:        Update actor $\pi^m$ for $\frac{\text{step}}{M}.p$ mini-batches (Eq. 5 and 8 or 9)      *Update actors*
22:      **else**
23:        Update actor $\pi^m$ for $\frac{\text{step}}{M}.p$ mini-batches (Eq. 2)
24:      **end if**
25:    **end for**

26:    **for** agent $m = 1\ldots M$ **do**
27:      $(\text{Fitness}_m, \_) \leftarrow \texttt{rollout}(\pi^m, \text{without noise, over } R \text{ episodes})$    *Evaluate actors*
28:    **end for**
29:    Rank population $\{\pi^m_{\phi,\theta}\}^M_{m=1}$ and identify top-$K$
30:    $\texttt{update\_archive}(\mathcal{G}, \{\pi^m_{\phi,\theta}\}^M_{m=1}, G)$
31: **end while**

---

COMPLEMENTARY PSEUDO-CODE FOR ARAC

Algorithms 2 and 3 respectively provide the pseudo-code of functions `rollout` and `update_archive` used in Algorithm 1.

---

**Algorithm 2** `rollout`

---

**Input:** actor $\pi$; noise status; number of episodes $E$; replay buffer $\mathcal{B}$;
Fitness $\leftarrow 0$
**for** episode $= 1, \ldots, E$ **do**
    $\mathbf{s} \leftarrow$ Initial state $\mathbf{s}_0$ from the environment
    **for** step $t = 0 \ldots$ termination **do**
        **if** with noise **then**
            Sample noise $z$
        **else**
            Set $z \leftarrow 0$
        **end if**
        $\mathbf{a}_t \sim \pi(.|\mathbf{s}_t, z)$
        Observe $\mathbf{s}_{t+1} \sim P(\cdot|\mathbf{s}_t, \mathbf{a}_t)$ and obtain reward $r_t$
        Fitness $\leftarrow$ Fitness $+ r_t$
        Store transition $(\mathbf{s}_t, \mathbf{a}_t, r_t, \mathbf{s}_{t+1})$ in $\mathcal{B}$
    **end for**
**end for**
Fitness $\leftarrow$ Fitness$/E$
**return** Average fitness per episode and number of steps performed

---

---

**Algorithm 3** `update_archive`

---

**Input:** archive $\mathcal{G}$; population of size $M$; maximal archive capacity $G$.
**if** $|\mathcal{G}| < G$ **then**
    Add all agents of current population to $\mathcal{G}$
**else**
    $c_1, c_2 \leftarrow$ 2-means(fitness of individuals in $\mathcal{G}$)
    **for** agent $m = 1, \ldots, M$ **do**
        Assign agent $m$ to closest cluster $c \in \{c_1, c_2\}$ based on its fitness
        Sample an archived agent $j \sim \text{Uniform}(c)$
        Replace archived individual $j$ by $m$
    **end for**
**end if**
**return** Updated archive $\mathcal{G}$

---

