# OpenReview forum: "Attraction-Repulsion Actor-Critic for Continuous Control Reinforcement Learning"
_ICLR.cc/2020/Conference — Reject_

### Official Review · AnonReviewer1 · 2019-10-08
**Official Blind Review #1**

**Rating:** 8

**Review:**

[Note to authors: I made profit of a small edit to add a new important comment, which starts with a star below]

This paper builds on SAC-NF, an extension of SAC with Normalizing Flows which is shown to improve exploration at a low cost (Mazoure et al., 2019). In this paper, the authors build a population of SAC-NF agents with different parameters for the NF, and use an attraction-repulsion approach to ensure diversity and performance of the population. The resulting combination is shown to be competitive to or better than state-of-the-art algorithms.

This is a nice paper with nice ideas, and the resulting ARAC algorithm shows state-of-the-art performance on difficult continuous control tasks. Thus I'm in favor of accepting it. However, it should be improved before final acceptance.

Here are a few random remarks:

About related work, "balancing the objective and diversity" is also the central concern of Quality-Diversity (QD) algorithms (see e.g. Cully&Demiris for a survey).
* Actually, in QD as well as Novelty Search (NS), the algorithms look for diversity in a so called "behavior space", which is also called "outcome space in Goal Exploration Processes (GEPs), see e.g. "O Sigaud, F Stulp (2108) Policy Search in Continuous Action Domains: an Overview, Neural Networks", Section 4 for a unifying view. According to eq. (8) and (9), ARAC is looking for diversity in "fitness space", which in my opinion is weaker. I would be glad to see a comment on that.

I had to have a look at (Rezende&Mohamed, 2015) and to read (Mazoure et al., 2019) to get the normalizing flow part. An effort could be made in the beginning of Section 2.3 to make the paper more self-contained.

The account of SAC corresponds to an early version of the algorithm. In the most recent one, the value function approximator has been removed (see "Soft actor-critic algorithms and applications").

I had a hard time to figure out whether the \beta of the NF had something to do with the \beta_\pi of the AR loss.

Fig. 4 is of much interest. I would be curious to see the performance of ARAC on Swimmer, as this benchmark has been shown to suffer from a deceptive gradient effect.

Reproducibility: "See github.com" => Can you be more specific ? :)

The caption of Fig 6 describes 3 things, but I can only see two curves.

Fig 7 seems to be a mere repetition of Fig 4. Late edit before submitting? ;)

Fig.8 suffers from a poor choice of colors: even magnifying a lot the pdf, I cannot tell which is TD3 and which is SAC.

Again, Fig.9 is the same as Fig. 2.

It would be nice to move Alg 1 to the main part if possible.

typos:

p3: without risk (of) loss of information

**Experience Assessment:**

I have published one or two papers in this area.

**Review Assessment: Checking Correctness Of Derivations And Theory:**

I assessed the sensibility of the derivations and theory.

**Review Assessment: Checking Correctness Of Experiments:**

I assessed the sensibility of the experiments.

**Review Assessment: Thoroughness In Paper Reading:**

I read the paper at least twice and used my best judgement in assessing the paper.

---

> ### Author Response · Authors · 2019-11-08
> **Answer to Reviewer #1**
>
>
> 1. Making paper self-contained for NF background:  Thank you for
> pointing this out, we will detail more this part.
>
> 2. Color of Fig. 8:  Indeed SAC and TD3 have similar color, we will change
> it in the next update.
>
> 3. Duplication of Figure 7 and 9:  We wanted to add a bigger version of
> these two plots in the appendix for better visualization

---

### Official Review · AnonReviewer3 · 2019-10-21
**Official Blind Review #3**

**Rating:** 3

**Review:**

The paper proposes an ensemble method for reinforcement learning in which the policy updates are modulated with a loss which encourages diversity among all experienced policies. It is a combination of SAC, normalizing flow policies, and an approach to diversity considered by Hong et al. (2018). The work seems rather incremental and the experiments have some methodological flaws. Specifically the main results (Fig. 4) are based on a comparison between 4 different codebases which makes it impossible to make meaningful conclusions as pointed out e.g. by [1]. The authors mention that their work is built on the work of Hong et al. (2018) yet the comparisons do not seem to include it as a baseline. I'm also concerned about how exactly are environment steps counted: in Algorithm 1 on line 27, it seems that the fitness which is used for training is evaluated by interacting with the environment yet these interactions are not counted towards total_step.

Further comments:
1. I don't agree with the first sentence in the abstract: there's nothing special about "continuous control" when it comes to deceptive rewards. Perhaps you should be more specific and refer to robotics-inspired environments or to "commonly used Mujoco environments".
2. I don't know what "non-convex continuous action spaces" refers to. All the environments studied in the paper have action space [-1, 1]^n which is a convex set.
3. I'm also not convinced that the existing environments have "deceptive" rewards--they were likely tuned so that learning on them is feasible.
4. The discussion regarding exploration completely ignores well studied methods based on e.g. upper confidence bounds or Thompson sampling. Unlike diversity based approaches, these methods are theoretically motivated.
5. In Section 2 you start with finite horizon MDPs but then present infinite horizon discounted set up.
6. I think the point about having only a single critic would deserve more discussion: what policy is this Q-value of? Would having an independent critic for each agent help or make things worse?

[1] Henderson et al. Deep Reinforcement Learning that Matters. (2017)

-------------------------------------------------------------------------------------------------------------------------------------------------------
Thanks for your comments. I'm still concerned about comparing to different codebases hence I'm keeping my score as is. Also it is still unclear to me whether you count the evaluation rollouts towards the total number of steps--fitness and hence learning is based on these rollouts so they should be included.

**Experience Assessment:**

I have published one or two papers in this area.

**Review Assessment: Checking Correctness Of Derivations And Theory:**

N/A

**Review Assessment: Checking Correctness Of Experiments:**

I assessed the sensibility of the experiments.

**Review Assessment: Thoroughness In Paper Reading:**

I made a quick assessment of this paper.

---

> ### Author Response · Authors · 2019-11-08
> **Answer to Reviewer #3**
>
> 1.Codebase for different baselines:
> Due to time constraints, we used the implementations of baselines as provided by the authors, which is a common practice in the field (see TD3, first paragraph of page 8; see Stabilizing Off-Policy Q-Learning via Bootstrapping Error Reduction, top of page 16).
>
> 2.Baselines comparisons: See answer 1 (b) to Reviewer 2.3.Environment step (x-axis): On the x-axis is reported the total number of environment steps meaning that 1 million environment steps on the x-axis represents roughly 200,000 environment steps for each agent (for a total of 5 agents).
>
> 4.Horizon of MDP: We apologize for the bad wording here. We wanted to emphasize that during evaluation time, the horizon is infinite (until the agent dies), while at training time, it is finite. See answer to Reviewer 2.3.
>
> 5.The phrasing of "continuous control rewards": The concern is fair, and we will update to "robotics control" as suggested.
>
> 6.The phrasing of "action space": This is a typo for which we apologize and which should just be "continuous action spaces". Paragraph 1 of the introduction has been updated.
>
> 7.Deceptive reward environments: Humanoid environments have a multimodal reward landscape (by the presence of a survival bonus) since there exist multiple suboptimal gait that can lead the agent to fall earlier (Conti et al, Improving Exploration in Evolution Strategies for Deep Reinforcement Learning via a Population of Novelty-Seeking Agents, 2018; Lehman et al, Novelty search and the problem with objectives, 2011), e.g. in humanoids, arm movement might or might not be synchronized with legs.
>
> 8.Concern about exploration method: We can understand your concerns about the method not being compared against pure exploration baselines. It is true that the word exploration can be used in different contexts. In our case, the focus is rather on efficient population-based optimization by avoiding local minima at training time.
>
> 9.Centralized critic: The goal of having a centralized critic is mainly to save computational cost. Moreover, having a centralized critic receive feedback only from the top-k members of the population allows to drive all agents towards the best solution seen so far.

---

### Official Review · AnonReviewer2 · 2019-10-23
**Official Blind Review #2**

**Rating:** 3

**Review:**

RL in environments with deceptive rewards can produce sub-optimal policies. To remedy this, the paper proposes a method for population-based exploration. Multiple actors, each parameterized with policies based on Normalizing Flows (radial contractions), are optimized over iterations using the off-policy SAC algorithm. To encourage diverse-exploration as well as high-performance, the SAC-policy-gradient is supplemented with gradient of an “attraction” or “repulsion” term, as defined using the KL-divergence of current policy to another policy from an online archive. When applying the KL-gradient, the authors find it crucial to only update the flow layers, and not the base Gaussian policy.

I have 2 issues with this paper:

1.	Lack of novelty – Most of the components of the proposed algorithm have been researched in other works. SAC with NF was contributed by Mazoure et. al., and this paper seems to use the exact same architecture.  The diversity-enforcing objective (Equation 1.) was proposed by Hong et. al., albeit not with a separate archive of policies, but using the experience replay (for off-policy) and recent policies (for on-policy). The objective is Section 3.2 is also the same as that in Hong et. al.

2.	I have major concerns about how the results have been reported:
     a.	The authors have omitted SAC and SAC-NF from Figure 4, and therefore the figure gives the impression that ARAC is significantly better than the presented baselines in some of the environments. For example, take Humanoid-v2. The original SAC paper reports reaching close to 5k score in about 0.5 million timesteps. It therefore seems that most of the benefit of ARAC (over the presented baselines) comes just from using SAC. Another example is Humanoid-sparse, where SAC-NF achieves ~550 (from Table 1), but is not shown in the figure, making ARAC (score ~600) look awesome. The authors also incorrectly mention in text that “ARAC is the only method that can achieve non-zero performance” on this.

     b.	Table 1 reports “maximum” average return. This is very misleading and non-standard in RL. Take Humanoid-sparse for instance. The number reported for ARAC is 816 which is the peak performance during training. As we see in Figure 4, ARAC is unstable and the perf. drops to ~600 at end of training. The peak performance during training is an irrelevant metric.

     c.	Humanoid-rllab range on y-axis looks incorrect


Minor points:

For creating a diverse archive, I’m not sure if k-means on the episodic-returns is the most effective mechanism. As explored in Conti et al., and also mentioned in the introduction of this paper, behavioral-diversity is a more useful characterization, and episodic-returns may not be aligned to that (especially in sparse reward environments).

Some of the missing related work (on population-based diversity): DIAYN, Learning Self-Imitating Diverse Policies, Divide and Conquer RL.


**Experience Assessment:**

I have published one or two papers in this area.

**Review Assessment: Checking Correctness Of Derivations And Theory:**

I assessed the sensibility of the derivations and theory.

**Review Assessment: Checking Correctness Of Experiments:**

I assessed the sensibility of the experiments.

**Review Assessment: Thoroughness In Paper Reading:**

I read the paper thoroughly.

---

> ### Author Response · Authors · 2019-11-08
> **Answer to Reviewer #2**
>
>
>
> 1.Lack of novelty:
>       (a)Differently from previous work, we explore in action space by using the expressivity of NF layers combined with AR coefficients from Hong (2018), but in the multi-agent setting.
>       (b)To highlight the benefit of NF layers, we adapted Hong’s method toSAC by operating AR only between Gaussian policies. Results in appendix Fig 6. show a drop in performance.
>       (c)Moreover, to quantify the effect of AR, we compare against 5 Independent agents: on Humanoid/Standup:5,300/145,000for Independent versus 6,000/163,000for ARAC. The combination of both elements is critical/essential and is the main contribution of our work.
>      (d)Finally, Hong (2018) restricted AR to the 5 most recent policies, while we maintain a diverse archive of ancestors (using clustering)and sample from it, as in Gupta (2012).
>
> 2.  (a)Regarding baselines: We wanted originally to compare against multi-agent methods in the main paper.  SAC and SAC-NF are both shown in the appendix figures to avoid clutter. Following your suggestion, we moved the comparison against single-agent baselines to the main manuscript.
>      (b)Max average return: We followed methodologies of other papers in the field. E.g. see TD3 (Fujimoto et al., Addressing FunctionApproximation Error in Actor-Critic Methods, 2018, see caption of Table 1 from TD3) and Trust-PCL (Nachum et al., Trust-PCL: AnOff-Policy Trust Region Method for Continuous Control, see CaptionTable 1 ). They suggest to first compute the average over N seeds(N=5 in our case) for every time step, and then reporting the maximal value of this series.1
>
> 3. Maximum timesteps in Mujoco:  By definition, all Mujoco environments have a maximum number of timesteps steps per rollout (variable "done" is set to True) as designed by Todorov et al. , 2012. However, this is not set for Humanoid-rllab (so the rollout during training stops when the agent reaches a terminal state). During evaluation time, this maximum number of steps is disabled and returns are collected until the agent dies. All baselines (ERL, CEM-TD3,CERL, TD3, SAC, SAC-NF) were also run under the above settings.
>
> 4.Concerns about diversity methods: Unlike options discovery methods (DIYAN, SECTAR, VIC, VALOR) which learn a set of diverse skills in an unsupervised learning manner, we tackle control tasks that require a singles kill by using extrinsic rewards. Quality Diversity (QD) doesn’t apply to non-homogeneous state components and large state spaces with features irrelevant to the downstream task (e.g. MuJoCo). We will clarify this in the next revision.
>
> 5.Archive buffer: The archive uses episodic returns as a proxy for policy diversity, since running time is of great concern in population-based meth-ods. One does not want to run behavioral diversity methods every couple of steps. We found that archive clustering is a cheap fix and a sufficient mean to ensure diversity in rewards. We ran an assessment of the archive fitness variance as a function of training steps for Ant, Humanoid and HumanoidStandup; empirical results suggest that clustering the fitness indeed does prevent the returns from collapsing onto a single mode (we added it to Appendix Fig. 6).

---

### Author Response · Authors · 2019-11-08
**General response to all reviewers**

We would like to thank all reviewers: the comments are very helpful. We believe we can address the concerns and comments well and improve the paper without deviating from the main story or results.

---

### Decision · Program_Chairs · 2019-12-19

**Decision:**

Reject

**Comment:**

The reviewers generally expressed considerable reservation about the novelty of the proposed method. After reading the reviews in detail and looking at the paper, I'm inclined to agree that the contribution is rather incremental. Using normalizing flows for representing policies in RL has been studied in a number of prior works, including with soft actor-critic, and I think the novelty in this work is limited in relation to prior work. Therefore, I cannot recommend acceptance at this time.